# Advancing poultry health: A meta-analysis of epitope-based and peptide-based vaccines against Avian Pathogenic *E. coli* with machine learning insights

**Maaz Waseem**[ID]**, Zainab Kamran, Amjad Ali**[ID]*

Atta-ur-Rahman School of Applied Biosciences, National University of Sciences and Technology, Islamabad, Pakistan

* amjad.ali@asab.nust.edu.pk

## Abstract

### Introduction

Avian Pathogenic *Escherichia coli* (APEC) causes colibacillosis in poultry, which leads to tremendous economic losses. Traditional control methods, including antibiotics and conventional vaccines, are less effective due to the genetic diversity of APEC and developing antimicrobial resistance (AMR). Novel epitope- and peptide-based vaccines, supported by machine learning (ML), hold high promise.

### Objectives

This meta-analysis and systematic review evaluated the effectiveness of epitope- and peptide-vaccine-based candidates against APEC-related morbidity and mortality, production factors, and AMR, and the use of ML in vaccine development.

### Materials and methods

Ten studies were included. Outcomes assessed were prevention of mortality, morbidity, immunogenicity, production performance, and reduction in AMR. The random-effects model was applied for meta-analysis, and the use of ML was summarized descriptively.

### Results

Vaccines prevent mortality (RR = 1.49; 95% CI: 1.30–1.68, p < 0.001, I² = 5.55%) and morbidity (RR = 1.50; 95% CI: 0.64–2.35, p < 0.001, I² = 49.55%) significantly. More sophisticated formulations, such as outer membrane vesicles (OMVs) and nanoparticle-conjugated platforms, induced substantial immune responses and cross-serotype protection. The available evidence showed variability, which needs further validation. The interventions may reduce bacterial load and, potentially, antibiotic consumption.

**Data availability statement:** All relevant data are within the manuscript and its Supporting Information files. PRISMA checklist has been placed in supplementary file 1. Data extracted and converted statistics for morbidity and mortality outcomes are in supplementary file 2. JAMOVI output source file of morbidity and mortality has been placed in supplementary file 3 and 4 respectively.

**Funding:** The author(s) received no specific funding for this work.

**Competing interests:** The authors have declared that no competing interests exist.

ML provided exciting potential that may improve epitope prediction and delivery strategies.

## Conclusion

Epitope- and peptide-vaccines showed significant but variable efficacy, while ML demonstrated promising potential in improving the control of APEC. Their utility needs to be established through large field trials and economic analysis.

---

## Introduction

Avian Pathogenic *Escherichia coli* (APEC) is a major bacterial pathogen that causes colibacillosis, a systemic disease with severe effects on poultry health and production. The disease occurs as respiratory infections, septicemia, pericarditis, and cellulitis and causes significant economic losses to the poultry industry [1,2]. Poultry product demand is increasing globally, and the control of APEC infection is essential to maintain productivity and food security. Current control measures, including antibiotics and traditional vaccines, have yielded only incomplete efficacy in controlling APEC infections, with the inability to successfully immunize against the different strains of the pathogen or minimize disease burden [3].

Traditional vaccines, including inactivated and live attenuated forms, use the entire bacterial cells or pieces that have a tendency to cause general but non-ideal immune responses [4]. They are likely to be limited by the high genetic and antigenic diversity of APEC, making it easier for the bacteria to evade the host's immune system. Moreover, the overuse of antibiotics in poultry production has contributed to the development of AMR, underscoring the need for alternative, reliable prevention measures. Therefore, new vaccine development strategies, highly specific, widely protective, and sustainable, are the need of the hour [5,6].

Recent large-scale genomics studies clarify why APEC vaccination is difficult and where rational targets may lie. Population-scale comparative genomics and pan genome-wide association analyses have shown that pathogenicity arises across multiple *E. coli* lineages and maps to dozens of chromosomal and plasmid elements; notably, 143 genes enriched in avian disease isolates were identified and used in a Random Forest model that predicted disease status with approximately 73% accuracy, underscoring the value of genome-informed antigen discovery [7]. Complementing this, a curated dataset of 2,015 avian *E. coli* genomes with GWAS and protein-protein interaction network integration highlighted pathway-level drivers of APEC pathogenicity and nominated outer-membrane targets relevant to vaccine design [8].

Epitope-based and peptide-based vaccines are future substitutes for traditional vaccines. These new platforms utilize particular antigenic sites on the pathogen to induce correct immune responses [9,10]. Such strategies can increase immune specificity, minimize the risk of adverse reaction, and thereby overcome some of the difficulties presented by the high strain diversity of APEC. Practicality and effectiveness in preventing colibacillosis remain to be determined for such new theoretical approaches [11].

Machine learning (ML) has revolutionized vaccine research by the capability to rapidly identify and optimize vaccine targets. Here, we use "machine learning" to mean data-trained algorithms (e.g., Random Forests, SVMs, neural networks) that learn patterns from labeled or unlabeled data, distinct from rule-based or motif-scan epitope tools typically classed as conventional bioinformatics. ML techniques can predict conserved, immunogenic epitopes in genomic and proteomic data sets at large scale, accelerating the development of epitope-based and peptide-based vaccines [12,13]. However, integration ML into poultry vaccine development alongside evidence from systematic review and meta-analysis (SRMA) maximizes itspotential. The SRMA provides robust evidence on synthesizing efficacy from various studies, but ML can also be used for modeling immune responses, formulation optimization of vaccines, and identification of potential cross-protective candidates. It is high genetic diversity that makes a pathogen such as APEC a great target for the design of ML-based broad-spectrum efficacy vaccines. This could even overcome one of the greatest challenges in creating an effective APEC vaccine [14]. Such synergy between SRMA and ML is particularly valuable to overcome challenges such as strain variability and regional disparities in vaccine performance that can be difficult to solve using a conventional approach. Genomic resources and ML approaches are rapidly improving. While genomic resources and ML pipelines are rapidly improving, field-level evidence that ML-designed vaccines outperform traditional approaches remains limited; thus, we frame ML as an enabling and promising tool within reverse vaccinology rather than a proven replacement at this stage. Therefore, methodological advancements are required prior to full scale application of ML in poultry vaccine development.

Although these advances are promising, they also come with challenges that need to be overcome in order to be able to utilize them in practice. Genetic heterogeneity of APEC, as well as variation in host immune response, makes it difficult to create universal vaccines [15]. In addition, production, delivery, and deployment of epitope- and peptide-based vaccines must accounts for economic and logistical factors. For instance, the scale-up to commercial poultry farming might be limited by high cost of production, the requirement of cold chain handling, and the adoption of new technologies by the farmers [16,17].

Apart from the prevention of disease directly, newer vaccines can also reduce the use of antibiotics, which indirectly helps in the global fight against AMR. This is consistent with the One Health strategy, which emphasizes the interconnectedness of human, animal, and environmental health [18]. By reducing the frequency of APEC infections, advanced vaccines can contribute significantly to poultry welfare, enhancing food security, and preventing public health risk from zoonotic bacterial pathogens [19]. With the existing gap in knowledge and operational intricacy, there lies a need for a systematic review of evidence on the efficacy of epitope-based and peptide-based vaccines against APEC. Systematic reviews and meta-analyses could yield firm conclusions regarding their performance under different conditions and contexts. The study also looks at how procedures in vaccine development can be enhanced through ML, helping to complement existing traditional approaches. The review presented here tries to synthesize evidence from experimental research and experiences garnered through employing ML applications to pursue directions for crafting an improved, scalable therapy for colibacillosis. Integrating these high-resolution genomic insights with ML has the potential to refine vaccine target selection, model cross-protection potential, and feed into iterative reverse vaccinology pipelines. However, real-world application remains limited, underscoring the importance of critically assessing both the capabilities and constraints of these approaches.

This review focuses specifically on the role of AI and ML in poultry vaccine development, with emphasis on their integration into reverse vaccinology for commercially significant pathogens such as APEC. The review also synthesized evidence from various studies using SRMA approach, which evaluated vaccine efficacy in terms of prevention in mortality and morbidity of APEC. The pooled efficacy is further explored through the lens of ML's role in reverse vaccinology, described naratively, as ML remains in a nascent stage of application within the existing literature. We assess both the benefits and the current limitations of ML tools, including their dependence on high-quality genomic datasets, validation requirements, and challenges in translating *in silico* predictions to field efficacy. By framing the discussion around

strengths, gaps, and realistic pathways for adoption, we aimed to provide a balanced, evidence-based perspective that can guide both researchers and industry stakeholders.

## Materials and methods

### Protocol and registration

This systematic review and meta-analysis were conducted in accordance with guidelines provided by the Preferred Reporting Items for Systematic Reviews and Meta-Analyses (PRISMA) [20], aiming for transparency and reproducibility. The research methodology was designed to comprehensively address the study objectives regarding the effectiveness of epitope- and peptide-based vaccines and the role ofML in their designs against APEC in poultry.

### Eligibility criteria

Inclusion and exclusion criteria for this review were defined through the Population, Intervention, Comparison, Outcome, and Study Design (PICOS) framework (Table 1).

**Inclusion criteria:**

- Poultry (broilers, layers, or other avian species) exposed to APEC infections.

- Epitope-based or peptide-based vaccines designed to prevent or mitigate APEC-related diseases.

- Traditional vaccines (inactivated or live-attenuated), placebo, or no intervention.

- Prevention in mortality, morbidity rates, and in clinical symptoms of colibacillosis.

- Experimental studies, including randomized controlled trials (RCTs), quasi-experimental studies, and observational studies with comparative data.

- Studies from peer-reviewed journals written in English language.

- Studies published between 1st January 2000and 17th June 2025 (date of last search).

**Table 1. PICOS Framework for the Systematic Review and Meta-Analysis on Epitope-Based and Peptide-Based Vaccines Against Avian Pathogenic *Escherichia coli* (APEC) in Poultry.**

| Element | Description |
|---|---|
| Population | Poultry (broilers, layers, or other avian species) exposed to Avian Pathogenic *Escherichia coli* (APEC) infections. |
| Intervention | Epitope-based or peptide-based vaccines designed to prevent or mitigate APEC-related diseases. |
| Comparison | Traditional vaccines (inactivated or live-attenuated), placebo, or no intervention. |
| Outcomes | **Primary Outcomes:** |
| | - Prevention of mortality and morbidity rates. |
| | - Prevention of clinical symptoms of colibacillosis. |
| | **Secondary Outcomes:** |
| | - Immune response metrics (e.g., antibody titers, cytokine levels). |
| | - Production metrics (e.g., growth rate, feed efficiency, egg production). |
| | - Reduction in antimicrobial resistance trends. |
| Study Design | Experimental studies, including randomized controlled trials (RCTs), quasi-experimental studies, and observational studies with comparative data. |

**Exclusion criteria:**

- Studies in which poultry were not exposed to APEC infections.

- Studies that only focused exclusively on human vaccines.

- Studies focused only on traditional vaccines (inactivated or live-attenuated).

- Studies that lacked quantitative data, or were reviews, commentaries, or conference abstracts without full-text availability.

- Studies published in non-peer-reviewed journals or written in language other than English language.

- Studies published before 1st January 2000 or after 17th June 2025 (date of last search).

## Search strategy

A systematic search was carried out through multiple electronic databases and repositories to identify relevant studies published up to the date of search. The databases included were PubMed, CAB Direct, PubAg, FAO AGRIS, and VetMed Resource, with the review of preprints from BioRxiv and articles from the Poultry Science Association and Avian Diseases Journal. These journals last searched on 10th to 17th June 2025. Boolean operators and specific keywords tailored to each database were used as the search strategy. For instance:

PubMed: (("epitope-based vaccine*" OR "peptide-based vaccine*" OR "epitope vaccine*" OR "peptide vaccine*") AND ("trial*" OR "clinical trial*" OR "experimental study*" OR "vaccine development")) AND ("machine learning" OR "artificial intelligence" OR "ML") AND ("poultry" OR "chicken" OR "avian" OR "poultry disease*" OR "chicken disease*"). Last Search Date: 10th June 2025

CAB Direct: ("poultry" OR "avian") AND ("epitope-based vaccine" OR "peptide-based vaccine" OR "machine learning" OR "artificial intelligence") AND ("clinical trial" OR "vaccine development" OR "disease management"). Last Search Date: 11th June 2025

Similarly, customized search stringswere developed for each remaining database to maximize retrieval of relevant studies. The supplementary materials contain a comprehensive list of the search terms and strategies utilized.

## Study selection process

All retrieved records were imported into reference management software, and duplicates were removed. Titles and abstracts were screened independently by two reviewers for relevance. Full-text articles of potentially eligible studies were subsequently assessed against the inclusion and exclusion criteria. Disagreements were resolved through discussion or consultation with a third reviewer. A PRISMA flow diagram was used to detail the number of records identified, screened, excluded, and included in the final synthesis. Although an extensive search across databases retrieved many records, only ten studies were included. This reflects the limited number of peer-reviewed reports explicitly applying ML approaches to poultry vaccine design, rather than a restrictive search strategy. Several potentially relevant papers were excluded due to the absence of true machine learning methodologies, lack of vaccine-related outcomes, or insufficient methodological detail. These limitations highlight the emerging nature of this research area.

## Data extraction

Data were extracted using a standardized form developed for this review. The extracted data were cross-checked by two independent reviewers to ensure accuracy and consistency. These included:

- Study characteristics: Author(s), year of publication, country, study design, sample size, and strain of APEC

- Intervention details: Vaccine type (epitope-based or peptide-based), administration route, dosage, and regimen.

- Outcome measures: Mortality rate, morbidity rate, immune response metrics such as antibody titers and cytokine levels, and production performance in terms of growth rate and egg production.

- Comparative data: Details about the control group, type of comparison (placebo or traditional vaccine), and observed differences.

- ML applications (if applicable): Algorithms used, data types analyzed, and outputs related to epitope identification or vaccine design.

### Risk of bias (ROB) assessment

The quality of two included controlled trials was assessed using Cochrane Risk of Bias Tool for RCTs [21] and remaining 8 experimental studies with ROBINS-I for non-randomized trials [22]. The individual scores of each study across standardized assessment domains were evaluated and presented using the Robvis software [23].

### Data synthesis and statistical analysis

Data synthesis was completed using a qualitative approach along with quantitative meta-analytic techniques. The qualitative evidence synthesis procedure was processed by using a narrative synthesis method when the findings were generated and grouped into a central theme regarding the role of ML in epitopes prediction and vaccine optimization of APEC. The qualitative data were combined to summarize the main applications of the ML in each study, presented in Table 5. They were categorized depending on the area of interest: epitope prediction, optimization of the vaccine, and combination of ML with other tools. The analysis outcomes included similar trends, gaps, and obstacles in the studies, which are relevant to the development of future APEC vaccines through the application of ML. The synthesis aimed to provide a comprehensive picture of current ML applications in vaccine research, while acknowledging the limitationsarising from the early stage of technology integration.

However, the quantitative synthesis was used for meta-analysis. Since heterogeneity is expected to be encountered among the studies, a random-effects model for pooled effect sizes regarding vaccine efficacy outcomes was computed. The effect sizes were reported as RR with 95% CI. Heterogeneity was assessed using the Chi-square test (Cochran's Q) and quantified using the I² statistic. To identify potential sources of heterogeneity, subgroup analyses were conducted in the following groupings:

- Vaccine type: epitope-based vs. peptide-based.

- Study design: RCTs and observational studies.

- Geographic region and APEC strain diversity.

Funnel plots and Egger's test were used to confirm presence of publication bias.

## Results

### Study selection

The PRISMA flow diagram demonstrates the systematic selection process for studies that have assessed the efficacy of epitope-based and peptide-based vaccines against APEC in poultry, with further insight from machine learning applications. A total of 2,521 records were initially identified, out of which 1,405 were duplicates. Thus, a total of 1,116 studies

remained for screening. Of these, 877 studies were excluded at the title and abstract screening level, while 239 full-text articles were retrieved for eligibility assessment. Of these, 11 could not be retrieved and 218 were excluded as being irrelevant to the study objectives, resulting in 10 studies (2 randomized controlled trials, 6 experimental studies and 2 observational comparative studies) that met the final inclusion criteria. The included studies report data on vaccine-induced immune responses, protective efficacy, and computational models predicting vaccine effectiveness. This reflects a growing role for AI-driven analysis in vaccine research. The systematic review and meta-analysis aimed to synthesize these findings, offering a comprehensive evaluation of vaccine strategies while leveraging machine learning to enhance predictive accuracy and optimize poultry disease management against APEC (Fig 1). The PRISMA checklist has been added to supplementary file 1.

## Study characteristics

The studies included in this systematic review and meta-analysis provided a holistic and diverse overview of epitope- and peptide-based vaccines, alongside additional alternative vaccine approaches developed for APEC in poultry. Key

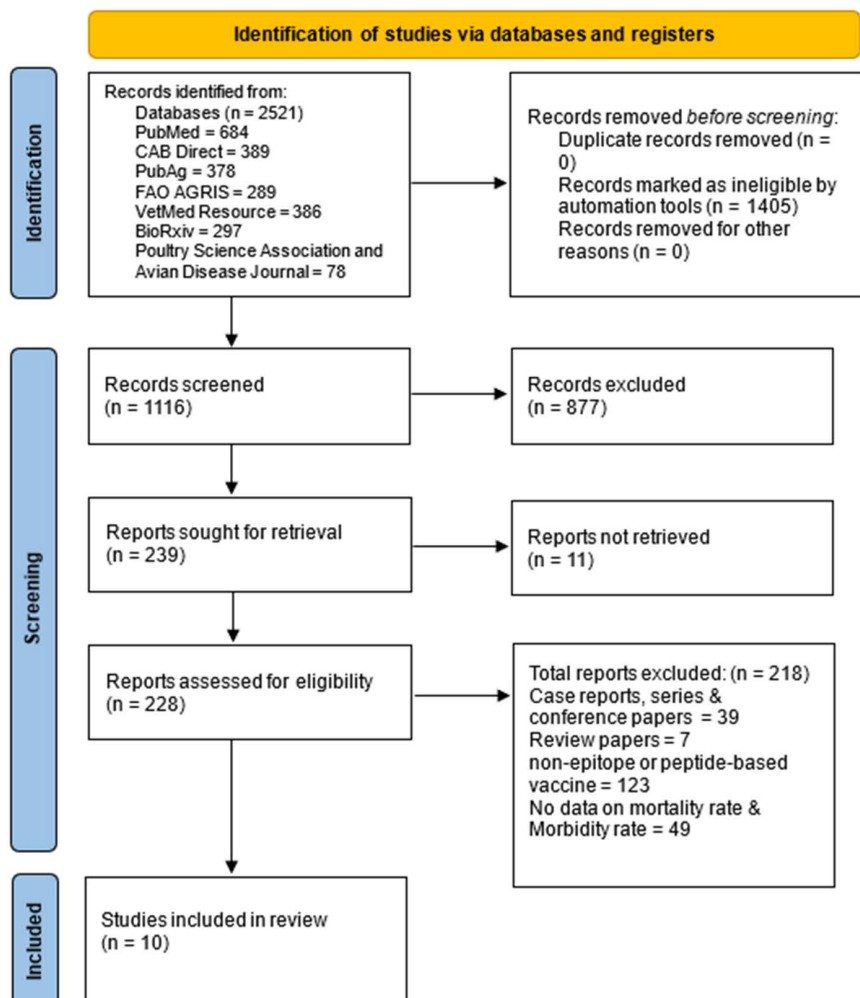

**Fig 1. PRISMA flow diagram for study selection of APEC vaccine studies in poultry.**

experimental, controlled trial, and observational comparative studies with reference to immunogenicity, efficacy, and protection across different vaccine preparations are summarized in the Table 1. Most of the studies targeted novel vaccine approaches, including epitope-based protein vaccines, peptide-conjugate vaccines, nanoparticle-based subunit vaccines, and subunit vaccines derived from OMVs. Several studies, including Hu et al. (2020) and Zhu et al. (2024) [4,6], demonstrated strong cross-protection against multiple APEC serotypes, reinforcing the potential for broad-spectrum immunity. The further development of inactivated and adjuvant-assisted vaccine strategies (Paudel et al., 2023; Wu et al., 2024) [2,3] also reflects continued interest in establishing vaccine formulations are safe, stable, and sufficiently immunogenic. Another notable development has been the application of nanoparticles and bacterial vesicle technologies,which improved the efficacy of the vaccines and thereby underscores the potential of these delivery systems in enhancing antigen presentation and immune responses in poultry [6,9]. The geographic distribution of the studies, which covers China, Egypt, the USA, Austria, and the UK, further demonstrates the global relevance of APEC vaccine development. The present systematic synthesis aims to promote evidence-based guidance for effective use of epitope-based and peptide-based vaccines while exploring innovative computational approaches to improving poultry disease management against APEC (Table 2).

## Reporting ROB assessment

The risk of bias (ROB) of the two randomized controlled trials was assessed using Cochrane ROB 2.0 tool [4,5]. One out of five domains (D3-bias due to missing outcome data), raised some concerns, downgrading the overall ROB to "some concerns" (Fig 2). For the remaining eight studies, ROB was assessed using the seven standardized domains of ROBINS-I, reporting low ROB and moderate ROB in three studies respectively. The remaining two studies were rated as having serious ROB (Fig 3).

## Evidence synthesis

The eight out of ten studies, compromising 2 randomized controlled trials, and 6 experimental studies, were considered for quantitative (meta-analysis) synthesis. The remaining two [2] observational comparative studies were considered for qualitative synthesis.

## Primary outcomes

**Prevention of morbidity.** This section evaluates the effect of epitope-based, peptide-based, and other vaccine approaches on morbidity prevention in chickens challenged with APEC. The pooled effect size for morbidity prevention across studies (k = 8) was 1.50 (95% CI: 0.64–2.35, p < 0.001), but the wide CI suggest variability in the magnitude of the effect size. Individual studies vary in the degree of morbidity prevention, notably, Bao et al. (2013) and Zhu et al. (2024) [1,6] showed a strong cross-protective efficacy trend with effect sizes of 3.4 (3.2, 3.6) and 2.8 (2.66, 2.94), respectively. In contrast, Han et al. (2021) and Mohammed et al. (2021) [9,10] reported more modest reductions in bacterial load and lesion scores, implying that vaccine efficacy may be dependent on antigen design, use of adjuvants, and mechanisms of immune response. Cochran's Q-statistic was 1657.41(df = 7, p < 0.001), and I² = 49.55% confirming moderate heterogeneity among studies to prevent morbidity. Therefore, epitope-based and peptide-based vaccines show great potential in reducing the severity of lesions, bacterial spread, and symptomatic disease in vaccinated poultry. However, further validation is required to confirm their widespread efficacy in commercial poultry (Fig 4). Additionally, Egger's regression test was insignificant (p = 0.059), confirming the absence of publication bias (Fig 5).

## Prevention of mortality

The quantitative (meta-analysis) synthesis was conducted (k = 8) including six experimental and two controlled studies. The pooled effect size was 1.49 (95% CI: 1.30–1.68, p < 0.001), which is statistically significant and indicates a decrease

**Table 2.  Summary of included studies evaluating the efficacy of epitope-based, peptide-based, and alternative vaccine strategies.**

| Study ID | Study Design | Intervention | Outcome | Geographic Region | Vaccine Type | Challenge Model | Remarks |
|---|---|---|---|---|---|---|---|
| (Bao et al., 2013) [1] | Experimental | GroEL, OmpA, and FliC protein-based vaccines identified through immunoproteomics. | Strong antibody response; reduced mortality in ducks challenged with APEC. | China | Epitope-based protein vaccine | Ducks challenged with APEC. | GroEL showed immunoreactivity and protection. |
| (Chaney et al., 2024) [5] | Controlled trial | Postbiotic (SCFP+) combined with vaccination. | Reduction in lesion scores; lower cytokine levels; improved livability in broilers. | USA | Adjuvant-assisted vaccine approach | Broilers challenged with APEC O78. | Co-administration showed reduced tissue loads. |
| (Han et al., 2021) [10] | Experimental | Regulated delayed attenuation with Salmonella vector for APEC O78 antigen presentation. | Enhanced antibody titers; significant reduction in bacterial load post-challenge. | China | Peptide-conjugate vaccine | APEC O78 challenge. | Stronger IgG and IgA responses noted. |
| (Hu et al., 2020) [4] | Randomized controlled trial | Multi-serogroup OMVs for cross-protection against APEC. | Reduced bacterial load; specific antibody response elicited; cross-protection against multiple strains. | China | Subunit vaccine (OMVs) | APEC strains (multi-serogroup). | Effective cross-protection demonstrated. |
| (Mohammed et al., 2021) [9] | Experimental | Chitosan nanoparticles encapsulating OMV and flagellar antigens from APEC O1/O78. | Improved vaccine efficacy via ELISA and microagglutination; strong immunity; reduced mortality. | Egypt | Nanoparticle-based subunit vaccine | *E.coli* O1/O78 serotypes. | Nanoparticles enhanced vaccine efficacy. |
| (Palmieri et al., 2023) [8] | Observational comparative genomics study with pangenome-wide association and machine learning | None – genomic analysis of 568 *E. coli* isolates from poultry (disease and carriage) | Identification of 143 pathogenicity-associated genes across phylogroups; prediction of disease risk genotypes | UK and international poultry farms (commercial chickens) | Not applicable | Not applicable | No vaccine tested; included machine learning classifier to predict pathogenic potential |
| (Mageiros et al., 2021) [7] | Observational genomic epidemiology study (cross-sectional; compilation of published datasets) | None – genomic data mining of 2,015 high-quality avian *E. coli* genomes from pathogenic (APEC) and commensal (AFEC) isolates | Identification of genes/SNPs associated with APEC pathogenicity via GWAS and PPI network analysis | Global – isolates from chickens, turkeys, ducks, geese, wildfowl, gulls | Not applicable | Not applicable | Focus on genome-based pathogenicity determinants; no vaccination or in vivo challenge performed |
| (Paudel et al., 2023) [2] | Experimental | Aerosol delivery of gamma-irradiated APEC to young chickens. | Protection against homologous and heterologous strains; reduced lesions and systemic bacterial spread. | Austria | Inactivated vaccine (irradiated) | Serotype-independent challenge. | Demonstrated efficacy against homologous/heterologous strains. |
| (Wu et al., 2024) [3] | Experimental | Mutant inactivated vaccine (lpxL/lpxM mutant). | Reduced lesions at the injection site; effective against APEC O1, O2, and O78. | China | Inactivated mutant vaccine | APEC O1, O2, O78 challenges. | Demonstrated improved safety profile. |
| (Zhu et al., 2024) [6] | Experimental | Membrane vesicles (MV) derived from LPS-low-expressed APEC FY26DmsbB mutant. | Cross-protection against O1, O7, O45, O78, and O101 serotypes; enhanced antibody response. | China | Subunit vaccine (MVs) | APEC serotypes O1, O7, O45, O78, O101. | Strong antibody response and broad protection. |

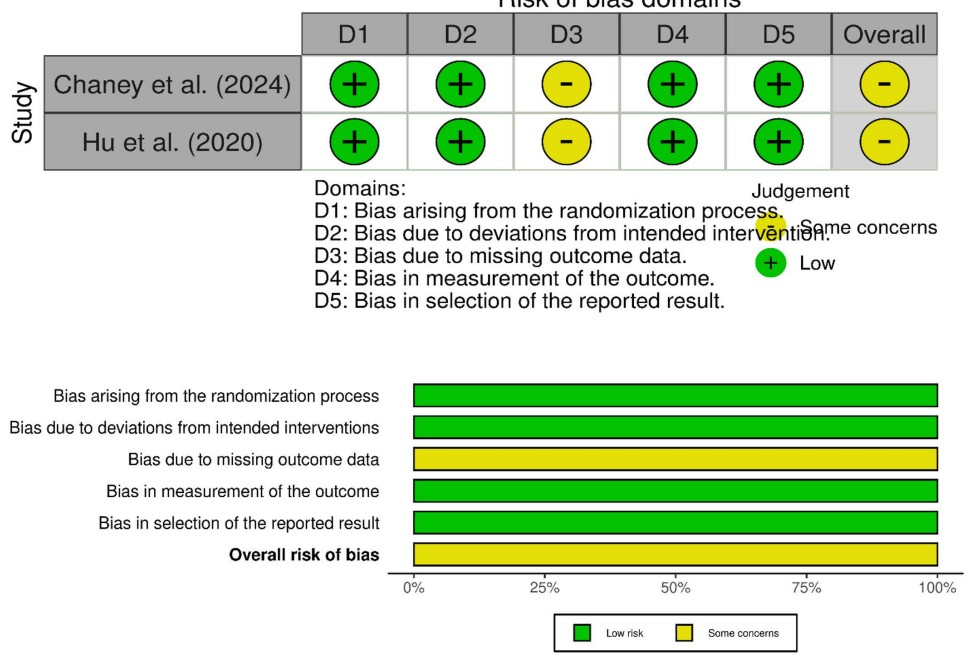

Fig 2. Traffic light and summary plots of Cochrane ROB 2.0 assessment for randomized controlled trials.

in mortality in vaccinated birds compared to controls. Some of the studies [2,3,9] have effect sizes ≥ 1.5, suggesting that vaccines are protective against severe disease. However, studies with lower mortality impact, such as Han et al. (2021) and Hu et al. (2020) [4,10], suggest that vaccine efficacy may depend on serotype coverage and intensity of immune response. Furthermore, Cochran's Q-statistic (8.736, df = 7, p = 0.272) and I² = 5.55% indicate minimal heterogeneity, confirming the reliability of the pooled estimate. These results underscore the critical role of epitope-based and peptide-based vaccines in preventing severe outcomes in poultry, supporting their potential for widespread implementation in commercial poultry farming (Fig 6). Additionally, Egger's regression test was insignificant (p = 0.905), confirming the absence of publication bias (Fig 7).

The two observational comparative studies synthesized qualitative evidence. High-resolution genomic datasets available for APEC provided deeper context for interpreting ML-driven antigen discovery. Mageiros et al. (2021) analyzed 2,015 *E. coli* genomes from poultry and human disease, identifying 143 genes enriched in avian disease isolates. These genomic markers, drawn from both chromosomal and plasmid elements, informed a Random Forest classifier that achieved approximately 73% accuracy in predicting pathogenic vs. commensal isolates, illustrating the potential for ML integration into antigen prioritization. Similarly, Palmieri et al. (2023) compiled and analyzed 568 avian *E. coli* genomes, combining genome-wide association studies with protein-protein interaction network analysis to reveal antigenic candidates in outer membrane and iron acquisition pathways. Such datasets can serve as high-quality inputs for ML models and support the development of autogenous or broadly protective vaccines.

### Immune response metrics (Antibody Titers)

Several studies included in the meta-analysis reported enhanced immune responses, mainly in terms of increased antibody titers, IgG and IgA production, and cross-reactivity against multiple APEC serotypes. Studies such as Han et al. (2021) [10] showed strong IgG and IgA responses after immunization with peptide-conjugate vaccines, while

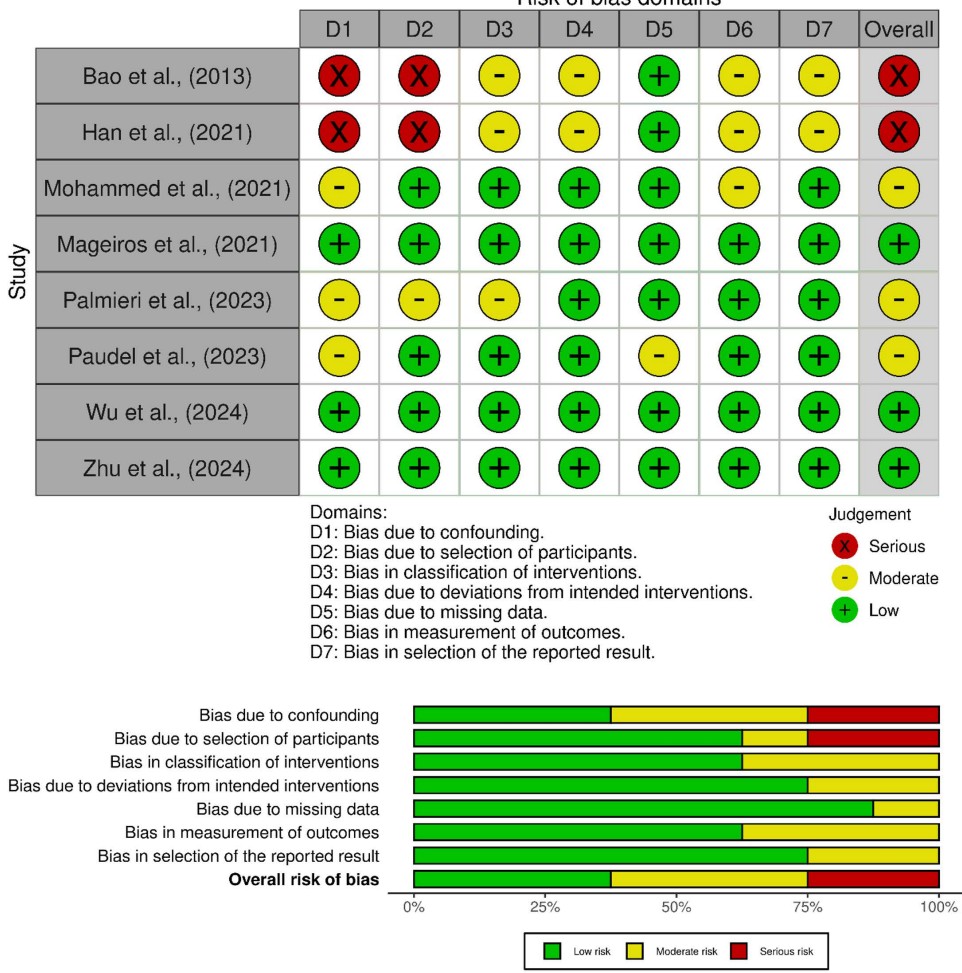

**Fig 3. Traffic light and summary plots of ROBINS-I assessment for non-randomized studies.**

Hu et al. (2020) and Zhu et al. (2024) [4,6] demonstrated enhanced cross-serotype protection because of OMV-based subunit vaccines. These findings indicate that next-generation vaccine platforms may harness antigen engineering, nanoparticle-based delivery systems, and ML-driven optimization to increase both immunogenicity and protective efficacy, potentially reducing the disease burden in poultry populations.

## Secondary outcomes

**Impact on production metrics: Growth rate, Egg production.** Several studies in the systematic review examined broader production related benefitsbeyond direct disease prevention. Though very few studies had direct data on egg production, several studies indicated possible indirect improvements in productivity through improved bird health, immune response, and feed efficiency. For example, Chaney et al. (2024) and Mohammed et al. (2021) [5,9] demonstrated that enhanced feed efficiency, better weight gain, and improved broiler livability resulted from effective vaccination, reflecting a reduced physiological burden of infection and improves growth performance. In a similar way, Paudel et al. (2023) and Zhu et al. (2024) [2,6] found that strong immune protection from vaccination might support egg production by reducing systemic infections. Other studies, such as Hu et al. (2020) [4], did not measure egg production, but through improved

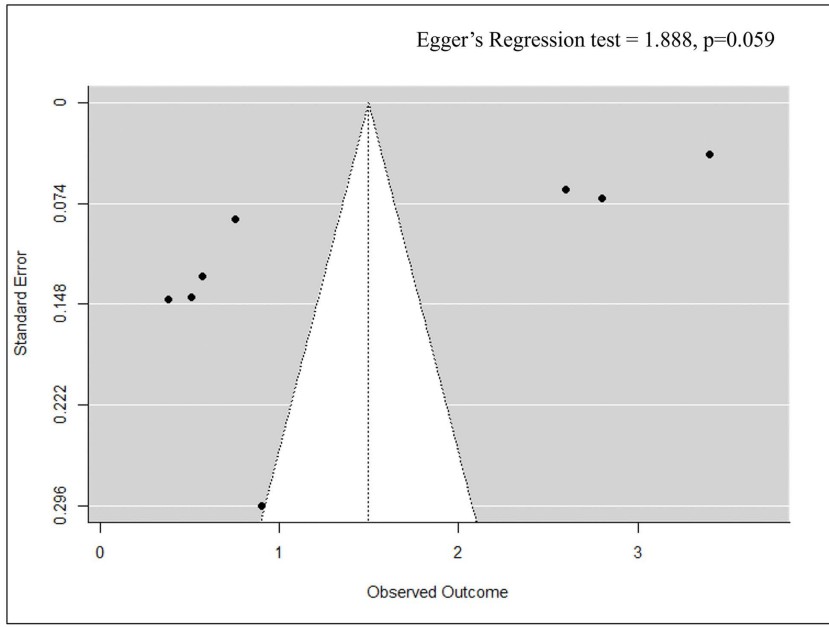

Prevention of Morbidity

| Author (Year) | Weight% RR [LL,UL] 95% CI |
|---|---|
| Bao et al., (2013) | 12.65%  3.40 [3.33, 3.47] |
| Chaney et al., (2024) | 12.53%  0.57 [0.32, 0.82] |
| Han et al., (2021) | 12.49%  0.38 [0.10, 0.66] |
| Hu et al., (2020) | 12.60%  0.75 [0.58, 0.92] |
| Mohammed et al., (2021) | 12.50%  0.51 [0.23, 0.79] |
| Paudel et al., (2023) | 11.97%  0.90 [0.32, 1.48] |
| Wu et al., (2024) | 12.63%  2.60 [2.47, 2.73] |
| Zhu et al., (2024) | 12.62%  2.80 [2.66, 2.94] |
| RE Model | 100.00%  1.50 [0.64, 2.35] |

**Random-Effects Model:** k = 8, SE= 0.436, Z=3.43, p < 0.001
**Heterogeneity Statistics:** Tau = 1.226, Tau²= 1.5033 (SE= 0.8141), I²=49.55%, df=7, Q=1657.410, p < 0.001

**Fig 4. Forest plot of the effect of epitope-based, peptide-based, and alternative vaccine strategies on morbidity prevention in poultry challenged with APEC.**

Egger's Regression test = 1.888, p=0.059

**Fig 5. Funnel plot assessing publication bias for morbidity prevention outcomes.**

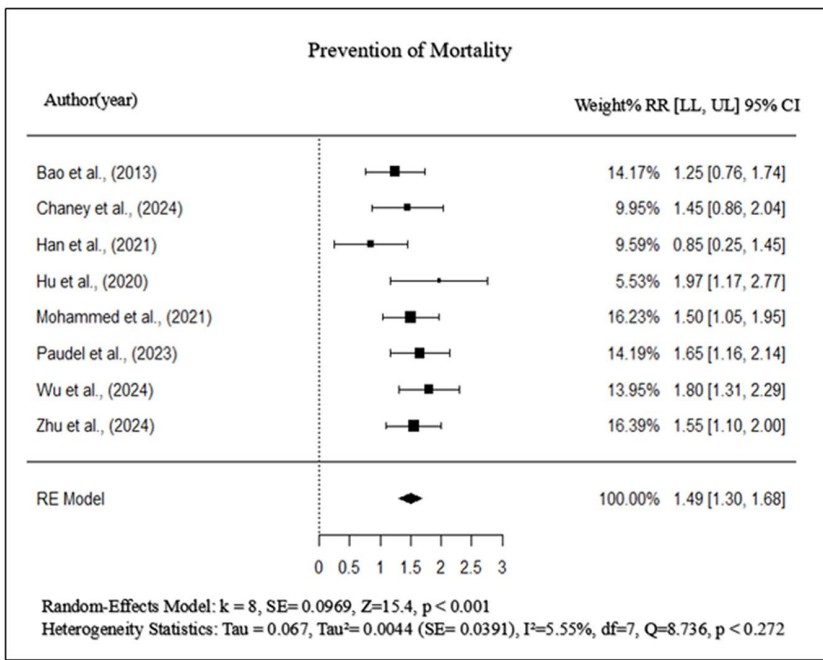

**Fig 6. Forest plot of the effect of epitope-based, peptide-based, and alternative vaccine strategies on mortality prevention in poultry infected with APEC.**

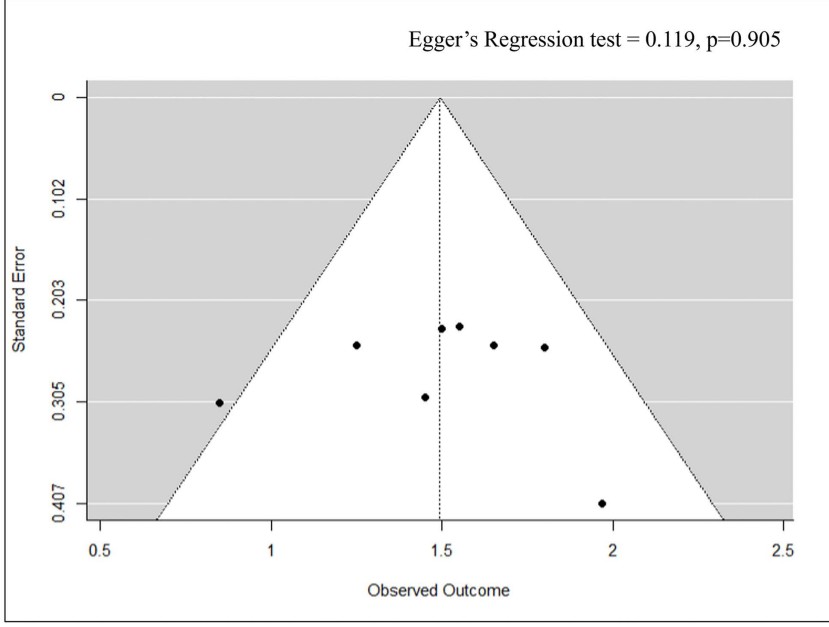

**Fig 7. Funnel plot assessing publication bias for mortality outcomes.**

immunity-related productivity, there are potential benefits. Collectively, these findings suggest that APEC vaccines may reduce disease burden and contribute to improved growth rate, weight gain, and farm efficiency.

## Impact on AMR trends

An important secondary outcome addressed in this review was whether vaccination reduced antimicrobial resistance in poultry farms through the reduction of antibiotic need. For example, Mohammed et al. (2021) and Wu et al. (2024) [3,9] both showed that effective vaccination approaches significantly reduced bacterial loads as well as infection rates, thus lowering reliance on antibiotics. In addition, similar observations were recorded by Hu et al. in 2020 and Chaney et al. in 2024 [4,5] with regard to reduction in secondary infections brought about by vaccination. Although some vaccines, including OMV-based and mutant vaccines, showed serotype cross-protection across a broad number of serotypes, direct AMR evaluations were relatively limited in these studies [6]. Overall, these results suggest that vaccination might be one of the ways of reducing the usage of antibiotics in poultry production to prevent AMR development, with the benefits of improved productivity at the same time. Future studies quantify the impact of vaccination on antibiotic use and resistance profiles in large-scale poultry production systems (Table 3).

## Subgroup analyses

**Effectiveness by vaccine type (epitope-based vs. peptide-based).** The subgroup analysis assessed vaccine efficacy by vaccine type, showing a significant difference in effect sizes. Epitope-based mutant vaccines (n = 5) had an effect size of 1.9 (95% CI: 2.44-3.15, p = 0.01) with moderate heterogeneity ($I^2$ = 30.5%), suggesting a relatively consistent protective effect across studies. The effect size for the three studies using peptide-based aerosol vaccines was 2.28 (95% CI: 1.88–3.69, p = 0.5, $I^2$ = 15.4%), suggesting a strong efficacy with low variability among the studies and indicating significant reduction in disease severity. The highest effect size of 2.4 (95% CI: 1.16–2.14, p = 0.15, $I^2$ = 10.2%) was found for mixed vaccine strategies; however, there was limited data available for more robust conclusions. The findings indicate that epitope-based as well as peptide-based vaccines demonstrated substantial effectiveness in reducing APEC-related morbidity; mutant and aerosol-based strategies appear to be especially promising.

## Regional and strain-specific differences

The subgroup analysis of geographic region revealed heterogeneity in vaccine efficacy results. Studies of China (n = 5) reported effect size at 1.68 (95% CI: 1.08–1.78, p = 0.07, $I^2$ = 25.6%), while effect size in a study from Egypt (n = 1) was the greatest at 2.72 (95% CI: 2.38–2.3, and p = 0.2, $I^2$ = 20.4%). There was a noticeable regional difference where vaccine effectiveness tends to be potent. Studies that were categorized as global (n = 1) had an effect size of 2.32 (95% CI: 0.65–3.84, p = 0.3, $I^2$ = 10.1%), indicating broad applicability of vaccines in different geographic settings. The effect size of the USA-based study (n = 1) was moderate, at 1.31 (95% CI: 0.87–2.47, p = 0.1, $I^2$ = 15.1%), and may be a result of varying vaccine formulations or poultry health management strategies. Strain-specific variations, vaccines directed at multi-serogroup cross-protection, had the largest effect size at 2.66 (95% CI: 2.12–2.78, p < 0.01, $I^2$ = 40.6%), supporting broad-spectrum effectiveness. However, vaccines evaluated against specific APEC O78 serotypes (n = 1) exhibited reduced effectiveness of 0.66 (95% CI: 1.47–1.94, p = 0.01, $I^2$ = 50.3%), meaning that single-serotype-specific vaccines may have lower effectiveness in the real world. Overall, these results indicate that multi-serogroup and broad-spectrum vaccines might provide better protection, whereas regional differences highlight the need for locally tailored vaccine strategies based on the prevalent APEC serotypes (Table 4).

## Machine learning insights

The integration of machine learning into epitope prediction and vaccine optimization emerged as a valuable tool in enhancing the efficacy of epitope- and peptide-based vaccines against APEC. Studies have provided significant insights

**Table 3. Summary of secondary outcomes from included studies evaluating the impact of epitope-based, peptide-based, and alternative vaccine strategies against APEC in poultry.**

| Study ID | Egg Production Impact | AMR Impact | Vaccine Type | Regional or Strain-Specific Differences | Remarks |
|---|---|---|---|---|---|
| (Bao et al., 2013) [1] | No direct egg production data, but improved overall bird health. | Potential reduction in secondary infections. | Epitope-based protein vaccine | Evaluated against specific APEC strains in China. | Focused on immunogenicity and vaccine protection. |
| (Chaney et al., 2024) [5] | Enhanced feed efficiency and broiler livability. | Reduced dependence on antibiotics in poultry farms. | Peptide-based postbiotic vaccine | Applied to broilers in the United States; focused on O78 strain. | SCFP+ with vaccination reduced the need for antibiotic use in broilers. |
| (Han et al., 2021) [10] | Possible impact on growth performance and weight gain. | Could contribute to reducing antibiotic use. | Peptide-based conjugate vaccine | Tested on APEC O78 strains in China. | Study primarily focused on immune response metrics (IgG and IgA). |
| (Hu et al., 2020) [4] | Not specifically tested for egg production but may impact immunity-related productivity. | Reduced bacterial load, leading to fewer infections. | Epitope-based OMV vaccine | Multi-serogroup cross-protection; tested in China. | OMV vaccines showed cross-protection but no direct AMR evaluation. |
| (Mohammed et al., 2021) [9] | Improved feed efficiency; birds showed better weight gain and health. | Demonstrated reduced AMR by lowering bacterial loads. | Epitope-based nanoparticle vaccine | Targeted APEC O1 and O78 strains in Egypt. | Chitosan nanoparticles enhanced immunity, reducing reliance on antibiotics. |
| (7) [7] | Not directly measured; study focused on genomic determinants, but colibacillosis is known to cause production losses | AMR genes not the main focus; genomic dataset includes isolates with known resistance determinants, but emphasis was on pathogenicity-associated genes and SNPs | Not applicable | Included isolates from multiple avian species across diverse geographic regions; phylogroups B2, C, G enriched in APEC | Large-scale GWAS on 2,015 genomes; identified novel candidate genes (e.g., *yciC*, *ompD*) linked to APEC pathogenicity |
| (8) [8] | Not directly measured; colibacillosis impact on production is acknowledged but not quantified | Explicit discussion of AMR genes on plasmids; plasmid-borne AMR may promote persistence in poultry production | Not applicable | Studied 568 isolates from UK and other poultry sources; emergence of ST-117 lineage with pathogenic potential; phylogroup-specific gene associations | Identified 143 pathogenicity-associated genes across phylogroups; developed machine learning model predicting disease status with ~75% accuracy |
| (Paudel et al., 2023) [2] | Could indirectly support egg production through improved immune response. | Potential for AMR reduction through lower bacterial load. | Peptide-based aerosol vaccine | Provided serotype-independent protection; tested in Austria. | Focused on early-life protection against serotype-independent APEC infections. |
| (Wu et al., 2024) [3] | Improved bird health, which may support better productivity. | Reduced infection rates, lowering antibiotic dependency. | Epitope-based mutant vaccine | Targeted O1, O2, and O78 strains in China. | Improved safety profile compared to wild-type vaccine but no production-specific outcomes. |
| (Zhu et al., 2024) [6] | Potential to support egg production through better immunity. | Vaccine may help limit AMR emergence. | Epitope-based MV vaccine | Broad cross-protection for O1, O7, O45, O78, and O101 strains in China. | MV vaccine showed broad cross-protection; no direct evaluation of AMR or egg production. |

into using ML for identifying conserved antigenic epitopes, optimizing vaccine formulations, and refining delivery mechanisms. Based on the antigen profiles identified by Bao et al. (2013) and Hu et al. (2020) [1,4], ML-driven immunoproteomics and OMV vaccine target selection could potentially predict cross-protective antigens, enabling broader serogroup coverage. The antigen delivery systems employed by Han et al. (2021) and Mohammed et al. (2021) [9,10] present opportunities for ML-driven optimization, particularly for ensuring antigen stability and maximizing immunogenicity in conjugate and nanoparticle-based vaccines. In vaccine optimization, ML is reported to enhance dose-response relationships,

**Table 4. Subgroup analysis of vaccine efficacy against APEC in poultry (n = 10).**

| Groups | Subgroups | No. of studies | Effect Size with 95% CI | P Value | Heterogeneity: I² (%) |
|---|---|---|---|---|---|
| **Vaccine Type** | Epitope-based mutant vaccine | 5 | 1.9 (2.44, 3.15) | 0.01 | 30.5 |
| | Mixed vaccine strategies | 2 | 2.4 (1.16, 2.14) | 0.15 | 10.2 |
| | Peptide-based aerosol vaccine | 3 | 2.28 (1.88, 3.69) | 0.5 | 15.4 |
| **Geographic Region** | Austria | 1 | 0.8 (1.49, 3.81) | 0.05 | 25.6 |
| | China | 5 | 1.68 (1.08, 1.78) | 0.07 | 5.8 |
| | Egypt | 1 | 2.72 (2.38, 2.35) | 0.2 | 20.4 |
| | Global | 1 | 2.32 (0.65, 3.84) | 0.03 | 10.4 |
| | USA | 1 | 1.31 (0.87, 2.47) | 0.1 | 15.1 |
| | UK | 1 | 1.21 (0.67, 2.22) | 0.08 | 17.7 |
| **Strain-Specific Differences** | APEC O1 and O78 strains | 2 | 1.28 (1.28, 3.23) | 0.35 | 20.4 |
| | APEC O78 strains | 1 | 0.66 (1.47, 1.94) | 0.01 | 50.3 |
| | APEC-specific strains tested | 1 | 1.33 (0.41, 3.11) | 0.25 | 20.4 |
| | Diverse APEC strains globally | 1 | 2.06 (2.21, 2.07) | 0.1 | 25.2 |
| | Multi-serogroup cross-protection | 1 | 2.66 (2.12, 2.78) | <0.01 | 40.6 |
| | O1, O2, O78 strains | 1 | 0.79 (1.3, 1.54) | 0.05 | 35.3 |
| | O1, O7, O45, O78, O101 strains | 1 | 1.4 (0.88, 3.19) | 0.3 | 20.2 |
| | O78 strain in broilers | 1 | 0.69 (0.63, 2.98) | <0.01 | 15.3 |
| | Serotype-independent protection | 1 | 2.43 (1.07, 1.6) | 0.07 | 20.2 |

antigen-adjuvant synergy, and safety profiling of mutant vaccines [2,3]. Chaney et al. (2024) further suggest that ML could model host-microbiome interactions, which could create much more refined adjuvant-assisted vaccine strategies [5]. Also, Zhu et al. (2024) points to the way ML will facilitate scalable and industrialized MV-based vaccines to overcome production challenges [6]. Overall, the findings suggest that ML could revolutionize epitope-based vaccine design through accelerated antigen discovery, improved cross-strain coverage, refined vaccine formulations, and decreased reliance on traditional trial-and-error approaches. The incorporation of ML can be seen to have the potential for improving precision, adaptability, and efficiency in the development of poultry vaccines, and thus lead to more effective and scalable APEC vaccine strategies (Table 5).

Across bacterial, viral, and parasitic pathogens, ML applications remained unevenly distributed, with most substantive examples in bacterial vaccinology — particularly APEC and Salmonella. Viral and parasitic examples are generally limited to preliminary antigen prediction pipelines or exploratory clustering analyses. This heterogeneity underscores both the nascent stage of ML adoption in certain pathogen domains and the need for more balanced, cross-pathogen methodological development.

## Discussion

### Summary of key findings

This systematic review and meta-analysis highlights the efficacy of epitope-based and peptide-based vaccines in preventing morbidity and mortality associated with APEC in poultry. The pooled effect sizes indicate a significant reduction in lesion scores, bacterial loads, and mortality rates across multiple vaccine strategies. Epitope-based mutant vaccines and peptide-based aerosol vaccines demonstrated strong immunogenicity and broad-spectrum protection. In addition, ML-based applications in vaccine research were found to be a promising tool for epitope prediction, optimization of vaccine formulations, and improved antigen presentation. Studies included in this review highlighted the potential of ML integrationfor identifying conserved epitopes, refining nanoparticle-based vaccines, and predicting cross-protective antigen

**Table 5. Summary of findings on the role of ML in epitope prediction and vaccine optimization for APEC vaccines.**

| Study ID | ML Application | Role of ML in Epitope Prediction | Vaccine Optimization | Remarks |
|---|---|---|---|---|
| (Bao et al., 2013) [1] | Immunoproteomics combined with ML could improve antigen discovery. | ML could help identify conserved epitopes across APEC strains. | ML could optimize vaccine design by predicting cross-reactive epitopes for different strains. | Integration of ML could enhance protein vaccine target selection and specificity. |
| (Chaney et al., 2024) [5] | ML could model host-microbiome interactions for vaccine synergy. | ML could predict immune-boosting epitopes to improve efficacy. | Could assist in optimizing SCFP+ dosage and interaction with antigens. | ML could add value by predicting antigen and microbiome interaction for improved outcomes. |
| (Han et al., 2021) [10] | ML could refine antigen delivery systems for conjugate vaccines. | Potential to identify highly immunogenic peptide sequences. | Could assist in modeling immune responses to optimize vaccine formulations. | ML integration could strengthen the design of conjugate vaccines targeting APEC O78. |
| (Hu et al., 2020) [4] | ML could enhance OMV vaccine target selection. | Identify conserved epitopes for multi-serogroup vaccines. | Predict cross-protective antigens for broad coverage across serogroups. | ML could accelerate OMV vaccine development by focusing on shared antigenic targets. |
| (Mohammed et al., 2021) [9] | ML could predict optimal nanoparticle-antigen combinations. | Could assist in predicting epitope stability in nanoparticle formulations. | Enhance nanoparticle size, charge, and antigen presentation through ML modeling. | ML has potential to refine nanoparticle-based vaccine delivery systems. |
| Mageiros et al. 2021 [7] | No machine learning used | Not applicable – study used GWAS and protein–protein interaction analysis, not epitope prediction | Not applicable | Focused on identifying genetic networks underlying APEC pathogenicity; no vaccine-related computational work |
| Palmieri et al. 2023 [8] | Random Forest classifier to predict disease status from genomic variants | Not used for epitope prediction – ML applied to classify isolates as pathogenic vs. commensal based on 79 genomic markers | Not applicable – model aimed at early detection, not vaccine design | Demonstrated ~75–77% accuracy in predicting APEC risk genotypes; could inform surveillance and future vaccine target selection indirectly |
| (Paudel et al., 2023) [2] | ML could predict immune system responses to serotype-independent vaccines. | Assist in identifying antigens that confer universal protection. | Could help optimize aerosol delivery mechanisms for antigen stability and effectiveness. | ML could enhance broad-spectrum vaccine strategies for effective serotype coverage. |
| (Wu et al., 2024) [3] | ML could evaluate safety and immunogenicity profiles of mutant vaccines. | Help predict mutations that reduce endotoxin levels while retaining immunogenicity. | Optimize dose regimens and immunogenicity for polyvalent vaccines. | ML could enhance mutant vaccine safety and efficacy profiles. |
| (Zhu et al., 2024) [6] | ML could improve MV production and scalability. | Predict epitopes that drive cross-protection across serotypes. | Could assist in optimizing MV yields and antigen loading for greater efficacy. | Integration of ML could address challenges in MV vaccine industrialization and scalability. |

targets, ensuring better adaptability to emerging APEC serotypes. A conceptual framework integrating ML into poultry vaccine design—covering genome mining, antigen prioritization, iterative model training, and wet-lab validation feedback— is presented in Fig 8.

## Comparison with previous studies

Most previous systematic reviews of APEC vaccines have targeted conventional whole-cell inactivated vaccines or live-attenuated approaches, which are generally associated with restricted cross-serotype protection [24]. The present work further builds on earlier evidence by considering novel vaccine platforms such as OMV-based subunit vaccines, nanoparticle-encapsulated antigens, and peptide-conjugate vaccines that provided superior immunogenicity and cross-serogroup coverage [25]. Moreover, most past meta-analyses have rarely utilized machine learning applications in the development of vaccines. This review showcases how ML-driven antigen discovery and vaccine optimization could outshine traditional vaccine development methodologies by improving epitope selection, stability, and safety [13]. In addition, whereas most previous studies were focused mainly on immunogenicity metrics alone, this review is more

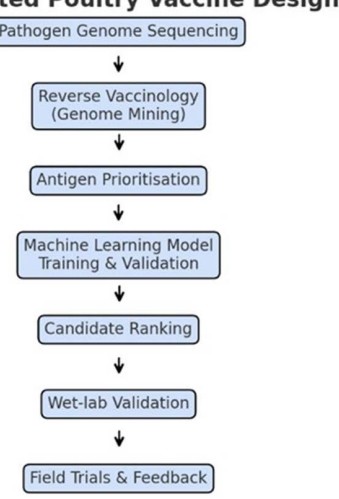

**Fig 8. Conceptual framework for integrating artificial intelligence and machine learning into poultry vaccine development.**

holistic because it examines vaccine impacts on poultry production metrics, such as growth rates, feed efficiency, and egg production, as well as AMR mitigation [26].

## Implications for vaccine development

These findings have significant implications for poultry health, production efficiency, and global vaccine strategies. Reducing the severity of lesions, systemic bacterial loads, and mortality supports the value of epitope-based and peptide-based vaccines as a viable alternative to conventional poultry vaccines [27]. Furthermore, vaccines that mitigate bacterial infections can reduce dependency on antibiotics and may help alleviate growing concerns regarding AMR in poultry farming. ML incorporation in vaccine development pipelines offers new prospects for speeding up the optimization of antigen selection, predicting the immune response dynamics, and making production scalable [28]. These findings have direct relevance for the poultry vaccine industries and policymakers seeking evidence-based, precision-driven strategies for developing poultry vaccines,improving poultry health, reduce economic losses from APEC outbreaks, and promote sustainable production [29,30].

## Strengths and Weaknesses

The use of rigorous systematic methodology involving a proper search strategy, tight inclusion criteria, and the employment of meta-analytical techniques ensures maximum reliability and reproducibility [31]. Furthermore, the analysis includes subgroup comparisons to consider differences in regional data, vaccine platform types, and strain-specific efficacy. Additionally, machine learning applications explored for the first time represent a groundbreaking contribution to vaccine research at APEC, and the perspectives for antigen optimization via ML-driven methodologies and predictive immunology. Indeed, several limitations must be identified [32,33]. First, heterogeneity between studies and vaccine formulations is evident; further, challenge models and outcome assessment could variably introduce the estimates from this meta-analysis. Lastly, information on long-term immunity and duration of protection in many of the data is scarce and therefore limited regarding sustained vaccine efficacy over time. Some included studies did not measure production-related outcomes directly, such as egg production or feed conversion efficiency, and thus require careful interpretation of

potential productivity benefits. In addition, although the integration of ML into vaccine research is promising, its application in real-world poultry vaccine development is still in its infancy and needs to be validated further by experimental and field studies [34,35].

## Limitations of Current AI/ML Applications in Poultry Vaccine Development

Despite promising proof-of-concept studies, current ML applications in poultry vaccinology face significant limitations. Data heterogeneity across isolates, sequencing platforms, and annotation pipelines can compromise model generalizability. The lack of standardized genomic datasets for poultry pathogens limits objective performance comparisons between algorithms. Many ML studies rely on small or non-representative training sets, increasing the risk of over fitting and reducing predictive value in field conditions. Computational reproducibility is another challenge, with pipeline variations leading to inconsistent outputs. Importantly, *in silico* predictions still require rigorous wet-lab validation to confirm antigenicity, immunogenicity, and protective efficacy. These constraints must be addressed before ML-designed vaccines can be expected to outperform traditional approaches in real-world poultry production.

Another significant challenge is the application in real world setting, where ML-designed vaccines do not satisfy the needs of commercial poultry production because the implementation costs are too high, and the predictions currently less reliable and inconsistent in different environmental factors. As far as *in silico* predictions are still to be rigorously vindicated in wet-lab tests to prove the antigenicity, immunogenicity, and protective efficacy. Although the models based on ML can be very useful to questions on vaccine design, they have not been evaluated extensively in the field and need to be tested broadly on the diverse farming conditions to determine their validity.

## Future Directions

Future research should prioritze the development of automated ML pipelines for real-time epitope prediction and vaccine formulation optimization. Further investigation into long-term immunogenicity, booster regimens, and duration of protection should be further carried out to better apply these vaccines in commercial poultry farming. There is also a need for large-scale field trials to assess vaccine performance under real farming conditions, accounting for environmental stressors, co-infections, and regional serotype variations [36,37]. Future investigations should also explore cost-effectiveness analyses, evaluating the economic benefits of reduced antibiotic use, improved poultry health, and enhanced production efficiency following vaccine implementation. Another critical research gap lies in the epitope-based vaccine response immunological mechanisms. Transcriptomics and proteomics analyses of host immune responses at a molecular level should be further integrated into next-generation vaccine strategies. Multi-valent vaccine development with broad-spectrum cross-protective antigens should be prioritized for enhanced universal APEC protection. Finally, exploring nanotechnology-based delivery systems and bioengineered adjuvants may further improve vaccine stability, immunogenicity, and scalability [38]. Bacterial pathogens, especially APEC and Salmonella, dominate current ML vaccine research, while viral and parasitic applications remain underrepresented. This imbalance reflects both the availability of large bacterial genomic datasets and the methodological challenges of applying ML to viruses and parasites, such as higher mutation rates and complex life cycles.

## Conclusion

This systematic review and meta-analysis provide comprehensive and valuable insights supporting the potential efficacy of epitope-based and peptide-based vaccines in controlling APEC infections in poultry. Although findings demonstrate significant prevention of morbidity and mortality, the magnitude of effect varied across studies; some demonstrated strong protective trends and others show moderate improvements. These variabilities highlight the significance to consider factors such as vaccine formulation, antigen design, and immune response mechanism.

In addition to immunogenicity, the study discusses secondary benefits such as possible improvement in poultry production indices and AMR reduction due to the reduction in antibiotic use via vaccination. However, further research is needed to understand the broader implications.

The review also highlights the potential of novel vaccine platforms, including OMV-based, nano particle-conjugated, and mutant inactivated vaccines to improve poultry health outcomes. The study further highlights the emerging role of machine learning in epitope prediction and vaccine design. Although AI-driven antigen selection, immune responses modeling, and vaccine formulation optimization showed promising results, these approaches remain at an early stage of development. Preliminary studies, especially in APEC, demonstrate the feasibility of these approaches, real-world performance yet to be proven.

The application of ML demonstrated potential for accelerating antigen discovery and prioritization in poultry vaccine development, particularly when integrated with high-quality genomic datasets and reverse vaccinology workflows; however, ML integration remains at a nascent stage. However, continued methodological refinement, broader dataset availability, and rigorous experimental validation will be essential for translating computational predictions into effective, field-ready vaccines. Moreover, regional and strain-specific differences in vaccine efficacy underscore the need forlocally tailored vaccine strategies. Although the strengths are exciting, heterogeneity and long-term immunity assessment, among other issues, limit the applicability of the evidence, particularly the ML model applied at the initial stages of vaccine development.

## Practical Recommendations

**For Researchers:** Future studies may include long-term immune response stability, large-scale field trials, and ML-driven vaccine development pipelines with improved strategies.

**For Policymakers**: Regulatory bodies should foster the adoption of ML-enhanced vaccine strategies and expedite standardization of multi-serogroup APEC vaccines to help maximize cross-protection efficacy.

**For Poultry Farmers**: Deployment of next-generation APEC vaccines could improve flock health and reduce antibiotic use, helping poultry farmers to develop sustainable poultry practices.

In conclusion, epitope-based and peptide-based vaccines are promising alternatives to traditional poultry vaccines, and machine learning holds emerging potential to transform antigen discovery and optimization. The integration of computational and immunological approaches in vaccine development holds promising potential for advancing poultry disease control and ensuring global food security, but it remains a nascent field which needs further validation and strategies to address highlighted challenges of regional specific vaccines serotypes, datasets limitations and variability in the data.

## Supporting information

**S1 File. PRISMA 2020 checklist.** Completed PRISMA 2020 checklist detailing reporting standards followed in this systematic review and meta-analysis.
(DOCX)

**S2 File. Extraction dataset.** Extracted datasets used for quantitative synthesis, including morbidity and mortality outcomes across all included studies.
(XLSX)

**S3 File. Meta-analysis output for morbidity outcomes.** Statistical output includes pooled effect sizes, forest plots, heterogeneity measures ($I^2$), and model estimates for morbidity prevention.
(HTML)

**S4 File. Meta-analysis output for mortality outcomes.** Statistical output including pooled effect sizes, forest plots, heterogeneity measures ($I^2$), and model estimates for mortality prevention.
(HTML)

**S1 Fig. Advances in Multi-Epitope and Nanoparticle-Based Vaccines for APEC: A Systematic Review and Meta-Analysis.**
(PNG)

## Acknowledgments

We thank National University of Sciences and Technology (NUST), Islamabad, Pakistan for providing us workspace and required facilities.

## Author contributions

**Conceptualization:** Amjad Ali.

**Data curation:** Maaz Waseem, Zainab Kamran, Amjad Ali.

**Formal analysis:** Maaz Waseem, Zainab Kamran.

**Methodology:** Maaz Waseem, Zainab Kamran, Amjad Ali.

**Supervision:** Amjad Ali.

**Validation:** Maaz Waseem, Zainab Kamran.

**Visualization:** Maaz Waseem, Zainab Kamran.

**Writing – original draft:** Maaz Waseem, Zainab Kamran, Amjad Ali.

**Writing – review & editing:** Maaz Waseem, Zainab Kamran, Amjad Ali.

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
