## [Decision Letter · Decision Letter 0]

1 Dec 2025

PONE-D-25-45349Advancing Poultry Health: A Meta-Analysis of Epitope-Based and Peptide-Based Vaccines Against Avian Pathogenic E. coli with Machine Learning InsightsPLOS ONE

Dear Dr. Ali,

Thank you for submitting your manuscript to PLOS ONE. After careful consideration, we feel that it has merit but does not fully meet PLOS ONE’s publication criteria as it currently stands. Therefore, we invite you to submit a revised version of the manuscript that addresses the points raised during the review process.

We look forward to receiving your revised manuscript.

Kind regards,

Arindam Mitra

Academic Editor

PLOS ONE

**Journal Requirements:**

1. When submitting your revision, we need you to address these additional requirements. Please ensure that your manuscript meets PLOS ONE's style requirements, including those for file naming. The PLOS ONE style templates can be found at https://journals.plos.org/plosone/s/file?id=wjVg/PLOSOne_formatting_sample_main_body.pdf and https://journals.plos.org/plosone/s/file?id=ba62/PLOSOne_formatting_sample_title_authors_affiliations.pdf 2. We note that your Data Availability Statement is currently as follows: All relevant data are within the manuscript and its Supporting Information files. Please confirm at this time whether or not your submission contains all raw data required to replicate the results of your study. Authors must share the “minimal data set” for their submission. PLOS defines the minimal data set to consist of the data required to replicate all study findings reported in the article, as well as related metadata and methods (https://journals.plos.org/plosone/s/data-availability#loc-minimal-data-set-definition). For example, authors should submit the following data: - The values behind the means, standard deviations and other measures reported;- The values used to build graphs;- The points extracted from images for analysis. Authors do not need to submit their entire data set if only a portion of the data was used in the reported study. If your submission does not contain these data, please either upload them as Supporting Information files or deposit them to a stable, public repository and provide us with the relevant URLs, DOIs, or accession numbers. For a list of recommended repositories, please see https://journals.plos.org/plosone/s/recommended-repositories. If there are ethical or legal restrictions on sharing a de-identified data set, please explain them in detail (e.g., data contain potentially sensitive information, data are owned by a third-party organization, etc.) and who has imposed them (e.g., an ethics committee). Please also provide contact information for a data access committee, ethics committee, or other institutional body to which data requests may be sent. If data are owned by a third party, please indicate how others may request data access. 3. PLOS requires an ORCID iD for the corresponding author in Editorial Manager on papers submitted after December 6th, 2016. Please ensure that you have an ORCID iD and that it is validated in Editorial Manager. To do this, go to ‘Update my Information’ (in the upper left-hand corner of the main menu), and click on the Fetch/Validate link next to the ORCID field. This will take you to the ORCID site and allow you to create a new iD or authenticate a pre-existing iD in Editorial Manager. 4. Please remove your figures from within your manuscript file, leaving only the individual TIFF/EPS image files, uploaded separately. These will be automatically included in the reviewers’ PDF. 5. If the reviewer comments include a recommendation to cite specific previously published works, please review and evaluate these publications to determine whether they are relevant and should be cited. There is no requirement to cite these works unless the editor has indicated otherwise.

**Additional Editor Comments:**

Please revise the manuscript in line with the comments of the reviewers in a point by point manner.

Reviewers' comments:

Reviewer's Responses to Questions

**Comments to the Author**

1. Is the manuscript technically sound, and do the data support the conclusions?

Reviewer #1: Partly

Reviewer #2: Yes

2. Has the statistical analysis been performed appropriately and rigorously? 

Reviewer #1: No

Reviewer #2: Yes

3. Have the authors made all data underlying the findings in their manuscript fully available?

Reviewer #1: Yes

Reviewer #2: Yes

4. Is the manuscript presented in an intelligible fashion and written in standard English?

Reviewer #1: Yes

Reviewer #2: Yes

5. Review Comments to the Author

**Reviewer #1:** The reviewer's comments are well-noted. Many similar topics exist in review articles on this subject. Therefore, the meta-analysis should emphasize which specific findings it contributes. The article seems suitable for publication regardless.

Please consider the following points for revision:

1. The manuscript provides a weak explanation of the fundamental definition of machine learning (ML) and its application in the study, particularly within the methods section. The authors mention using ML for the "Study Selection Process." However, the legality of this application for selection requires clarification. Using third-party application programming interfaces is generally acceptable, unlike certain generative AI tools currently prevalent. The description lacks clarity on whether the team developed the ML process internally, as it omits the specific algorithms used. The authors must confirm these details and remember that ML and AI have distinct definitions.

2. The authors must confirm and clarify the statement: "For the purposes of this review, 'machine learning' was defined as a data-driven approach in which an algorithm learns patterns from input data (e.g., supervised, unsupervised, or deep learning methods)." The description fails to mention the programming language or software tools used to build the data extraction process with ML.

3. The writing in the "Materials and Methods" section suggests possible use of AI for editing, leading to typographical errors like: "Each domain was assessed as "low risk," "high risk," or "unclear risk," " The authors should carefully review the entire section.

4. In this reviewer's opinion, the article presents a standard meta-analysis. The use of ML appears limited to automating selection and data extraction, while the core analysis techniques follow conventional meta-analysis approaches. The authors should confirm this point and adjust the title and "Materials and Methods" section to reflect accurately the role of ML and AI.

5. The manuscript repeats the full explanation for abbreviations, such as "outer membrane vesicles (OMVs)," multiple times. The authors should review and correct this pattern for all abbreviations used.

6. Figure 1, the PRISMA-P flowchart, should specify the reasons for excluding records during the screening. Eliminating 219 articles due to "different objectives" suggests a significant weakness in the initial keyword strategy. A more precise keyword selection would likely have prevented such a large-scale exclusion for that reason. The authors must provide a better explanation for these exclusions.

7. The abstract's graphical representation mentions vaccine effectiveness results, but the manuscript contains no corresponding forest plots. The reported parameters are also surprisingly limited to only two, raising questions about the difficulty of compiling data related to core vaccine effectiveness outcomes.

8. The manuscript must follow proper scientific nomenclature. For example, after the first full mention of "Escherichia coli," subsequent references should use the abbreviated form "E. coli." The authors should check the entire text for this issue.

The authors have no further comments but should perform a comprehensive check of the entire manuscript. The methods section requires significant strengthening. Please also double-check all author names and affiliations for accuracy, ensure reference citations are correct, and perform a final, thorough proofread.

**Reviewer #2:** Dear Authors,

Your manuscript addresses a timely topic with valuable insights on APEC vaccines and machine learning applications. Clarification of meta-analysis methodology, data reporting, and the specific contribution of machine learning would strengthen rigor, transparency, and scientific impact.

6. PLOS authors have the option to publish the peer review history of their article (what does this mean?). If published, this will include your full peer review and any attached files.

Reviewer #1: **Yes:** Mohammad Miftakhus Sholikin

Reviewer #2: No

---

## [Author Response · Author response to Decision Letter 1]

6 Feb 2026

Dear Editor,

We would like to thank staff of PlosOne and the reviewers for careful and thorough analysis of the manuscript. We found the feedback of your team and reviewers very constructive. We tried to be responsive to your concerns and want to extend our appreciation for taking the time and effort necessary to provide such insightful guidance.

Your comments and those of the reviewers were highly insightful and enabled us to greatly improve the quality of our research paper. Please find attached file, a revised version of our manuscript entitled “Advancing Poultry Health: A Meta-Analysis of Epitope-Based and Peptide-Based Vaccines Against Avian Pathogenic E. coli with Machine Learning Insights” (Manuscript ID: PONE-D-25-45349).

We hope that these revisions improve the overall quality of our write-up such that you and the reviewers now deem it worthy of publication in the PlosOne journal. Next, we offer brief responses to your comments as well as those of the reviewers.

The manuscript has been improved by considering your and reviewer’s valuable suggestions. We hope that you and the referees will find our responses satisfactory, and we are willing to include future suggestions.

Below, we respond to the comments by the referees, and indicate where relevant changes have been made. We look forward to working with you and the referees to move this manuscript closer to publication in the PlosOne journal.

In case of further queries, we will be happy to provide further details and look forward to your response.

Sincerely,

Amjad Ali, PhD

Peer review

The manuscript addresses an important topic concerning epitope- and peptide-based vaccines against APEC and provides a timely synthesis that integrates machine-learning applications. The subject is relevant to poultry health, AMR reduction, and vaccine innovation. The manuscript is generally well-structured and clearly written; however, several areas would benefit from clarification or methodological strengthening to enhance scientific rigor and transparency.

Response: We thank reviewers for taking time and constructively review our manuscript.

Major Comments

1. Scope and Positioning

While the review integrates ML as a major theme, the included studies do not uniformly apply ML. Clarifying the precise contribution of ML in the synthesis would improve conceptual coherence.

Response: We greatly appreciate the reviewer’s feedback, ML synthesis and impact in this manuscript are added at line # 227 to 237. Kindly review.

2. Methodology Transparency

The PRISMA methods are appropriately mentioned, but more detail is needed regarding search strings, inclusion/exclusion procedures, and the handling of AI-only narrative papers.

Response: We are thankful for your valuable suggestion. The search strings are updated in line # 178 to 185, inclusion/exclusion criteria is added in line # 147 to 168, and handling of AI only narrative papers in line # 227 to 237.

3. Meta-Analysis Details

Effect size values (RR > 1) appear to indicate increased risk rather than reduction unless inverted—clarification of directionality is recommended. Additionally, missing numeric data for several forest plots limits reproducibility.

Response: The wording has been inverted to prevention of mortality and morbidity to align interpretation with RR statistics. Kindly review meta-plots and figures with interpretation at line # 303 to 345.

4. Study Heterogeneity

Although heterogeneity is reported as low, differences in vaccine platforms, serotypes, species, and outcome measures should be acknowledged more explicitly.

Response: We are thankful for your valuable suggestion. The subgroup analysis is presented in table 4 and interpretation in line # 329 to 429. Kindly review.

5. Machine Learning Section

The ML component is promising but mostly descriptive. Strengthening this part with more concrete methodological criteria or limitations would improve the scientific contribution.

Response: We greatly appreciate the reviewer’s feedback. The methodological criteria is added in line # 227 to 237 and limitations of current AI/ML is already written and acknowledged in 518 to 528, but it is updated as per your suggestion in line # 529 to 536.

Comments

Title & Abstract

Clear and informative. The abstract would benefit from numerical context on included studies and a more cautious interpretation of ML findings.

Response: We thank the reviewer’s feedback. The revision has been made in the manuscript at line # 29 to 35.

Introduction

Well-written and comprehensive. A slightly tighter focus on the rationale for combining meta-analysis with ML discussion would increase coherence.

Response: The reviewer’s concern has been addressed in revised manuscript, the revision has been made on line # 87 to 90, 94 to 102, 129 to 133.

Materials & Methods

The PRISMA-based approach is appropriate; however, clarification on risk-of-bias scoring, data extraction consistency, and management of narrative ML-related information is recommended.

Response: The PRISMA scoring is added in methodology line # 220 to 225 and reported in result section 293 to 301. The plots have been added as figure 2,3 updated figures. Secondly, data extraction line # 202 to 214 and ML management of narrative is explained in data synthesis line 227 to 237. The below given references have been added in manuscript and highlighted in yellow.

• Higgins JP, Savović J, Page MJ, Elbers RG, Sterne JAC. Chapter 8: Assessing risk of bias in a randomized trial [last updated October 2019]. Available from: Chapter 8: Assessing risk of bias in a randomized trial | Cochrane

• Sterne JA, Hernán MA, Reeves BC, Savović J, Berkman ND, Viswanathan M, Henry D, Altman DG, Ansari MT, Boutron I, Carpenter JR. ROBINS-I: a tool for assessing risk of bias in non-randomised studies of interventions. bmj. 2016 Oct 12;355. https://doi.org/10.1136/bmj.i4919

Results

Results are clearly structured. Nevertheless, the reporting of effect sizes, heterogeneity, and subgroup analyses should be made more transparent, especially regarding exact study counts and numerical inputs.

Response: Respected reviewer, the meta-plots, subgroup analysis statistics and counts in PRISMA has been consistently reported and updated as per your suggestion. Kindly review the figure 1 to 7.

Discussion

The discussion provides valuable synthesis but could benefit from more balanced coverage of limitations, including dataset variability, model validation challenges, and real-world applicability of ML-based predictions.

Response: Respected reviewer, the revision has been made. kindly review lines 529 to 536.

Conclusion

Concise and aligned with findings. A more moderate tone regarding the transformative potential of ML would strengthen scientific neutrality.

Response: We are thankful for your valuable suggestion, the conclusion section has been updated and in the revised manuscript on line number 558 to 584.

Tables & Figures

Tables are informative. Forest plots and the PRISMA diagram should include complete numeric labels to improve reproducibility.

Response: Respected reviewer, all new plots and figures have been updated and added. Kindly see figure 2-7.

References

Comprehensive and up-to-date. Minor formatting inconsistencies can be corrected during editing.

Response: Respected reviewer, we have made improvements in the formatting. However, if there is further improvements.

Rebuttal Letter

Reviewer #1:

The reviewer's comments are well-noted. Many similar topics exist in review articles on this subject. Therefore, the meta-analysis should emphasize which specific findings it contributes. The article seems suitable for publication regardless.

Response: We thank reviewers for taking time and constructively review our manuscript.

Please consider the following points for revision:

1. The manuscript provides a weak explanation of the fundamental definition of machine learning (ML) and its application in the study, particularly within the methods section. The authors mention using ML for the "Study Selection Process." However, the legality of this application for selection requires clarification. Using third-party application programming interfaces is generally acceptable, unlike certain generative AI tools currently prevalent. The description lacks clarity on whether the team developed the ML process internally, as it omits the specific algorithms used. The authors must confirm these details and remember that ML and AI have distinct definitions.

Response: Respected reviewer, I have refined the methodology section as per your suggestion. The author’s intended meaning was that the ML role in vaccine development and designing among included studies. The statement has been omitted. The data extraction and literature search were performed as per PRISMA 2020 statement guidelines. However, the synthesis section has been updated by incorporating qualitative synthesis of the ML role (Line # 220 to 239).

2. The authors must confirm and clarify the statement: "For the purposes of this review, 'machine learning' was defined as a data-driven approach in which an algorithm learns patterns from input data (e.g., supervised, unsupervised, or deep learning methods)." The description fails to mention the programming language or software tools used to build the data extraction process with ML.

Response: The reviewer’s concern has been addressed in revised manuscript. We have omitted these sentences, and actual intended meaning was to present the definition of ML to synthesis evidence. The synthesis approach is now elaborated in line 220 to 230.

3. The writing in the "Materials and Methods" section suggests possible use of AI for editing, leading to typographical errors like: "Each domain was assessed as "low risk," "high risk," or "unclear risk," " The authors should carefully review the entire section.

Response: Respected Reviewer, the methodological quality assessment (risk of bias assessment) has been added Line # 213 to 218. Kindly review how it is conducted using Cochrane guidelines. The reporting is added in result section line # 285 to 292 along with figure 2,3.

4. In this reviewer's opinion, the article presents a standard meta-analysis. The use of ML appears limited to automating selection and data extraction, while the core analysis techniques follow conventional meta-analysis approaches. The authors should confirm this point and adjust the title and "Materials and Methods" section to reflect accurately the role of ML and AI.

Response: Respected reviewer, the ML role is identified using synthesis given in line #220 to 230. Yes, the core analysis is performed using conventional meta-analysis along with qualitative synthesis on ML role.

5. The manuscript repeats the full explanation for abbreviations, such as "outer membrane vesicles (OMVs)," multiple times. The authors should review and correct this pattern for all abbreviations used.

Response: The reviewer’s concern has been addressed in revised manuscript.

6. Figure 1, the PRISMA-P flowchart, should specify the reasons for excluding records during the screening. Eliminating 219 articles due to "different objectives" suggests a significant weakness in the initial keyword strategy. A more precise keyword selection would likely have prevented such a large-scale exclusion for that reason. The authors must provide a better explanation for these exclusions.

Response: Respected reviewer, the exact reasons for exclusion have been added now in Figure 1 (PRISMA Flowchart).

7. The abstract's graphical representation mentions vaccine effectiveness results, but the manuscript contains no corresponding forest plots. The reported parameters are also surprisingly limited to only two, raising questions about the difficulty of compiling data related to core vaccine effectiveness outcomes.

Response: The reviewer’s concern has been addressed in revised manuscript.

8. The manuscript must follow proper scientific nomenclature. For example, after the first full mention of "Escherichia coli," subsequent references should use the abbreviated form "E. coli." The authors should check the entire text for this issue.

Response: The reviewer’s concern has been addressed in revised manuscript

Reviewer #2:

Dear Authors,

Your manuscript addresses a timely topic with valuable insights on APEC vaccines and machine learning applications. Clarification of meta-analysis methodology, data reporting, and the specific contribution of machine learning would strengthen rigor, transparency, and scientific impact.

Response: We thank reviewers for taking time and constructively review our manuscript. The methodology has been updated with comprehensive and detailed explanation of quantitative synthesis (line #231 to 243) and qualitative synthesis (line # 219 to 230). The reporting of data extraction has been clarified in line # 199 to 212. The impact of ML insights has been clarified in methodology in Line #231 to 243 and in Table # 5. The forest plots and funnel plots have been updated and added from figure 4 to 7. The risk of bias assessment illustration is added as figure 2,3. Kindly review the revised version.

---

## [Decision Letter · Decision Letter 1]

26 Apr 2026

Advancing Poultry Health: A Meta-Analysis of Epitope-Based and Peptide-Based Vaccines Against Avian Pathogenic E. coli with Machine Learning Insights

PONE-D-25-45349R1

Dear Dr. Ali,

We’re pleased to inform you that your manuscript has been judged scientifically suitable for publication and will be formally accepted for publication once it meets all outstanding technical requirements.

Kind regards,

Yung-Fu Chang

Academic Editor

PLOS One

Additional Editor Comments (optional):

Reviewers' comments:

Reviewer's Responses to Questions

**Comments to the Author**

1. If the authors have adequately addressed your comments raised in a previous round of review and you feel that this manuscript is now acceptable for publication, you may indicate that here to bypass the “Comments to the Author” section, enter your conflict of interest statement in the “Confidential to Editor” section, and submit your "Accept" recommendation.

Reviewer #1: All comments have been addressed

2. Is the manuscript technically sound, and do the data support the conclusions?

Reviewer #1: Yes

3. Has the statistical analysis been performed appropriately and rigorously? 

Reviewer #1: Yes

4. Have the authors made all data underlying the findings in their manuscript fully available?

Reviewer #1: Yes

5. Is the manuscript presented in an intelligible fashion and written in standard English?

Reviewer #1: Yes

6. Review Comments to the Author

Reviewer #1: No additional comments are provided. The authors have adequately revised the manuscript. However, they should recheck the names, affiliations, and corresponding author to ensure accuracy and consistency. The authors should also carefully review the manuscript to confirm the scientific logic and conduct an independent proofreading of the entire text.

7. PLOS authors have the option to publish the peer review history of their article (what does this mean?). If published, this will include your full peer review and any attached files.

Reviewer #1: **Yes:** Mohammad Miftakhus Sholikin

---

## [Editor Report · Acceptance letter]

PONE-D-25-45349R1

PLOS One

Dear Dr. Ali,

I'm pleased to inform you that your manuscript has been deemed suitable for publication in PLOS One. Congratulations! Your manuscript is now being handed over to our production team.

Kind regards,

on behalf of

Dr. Yung-Fu Chang

Academic Editor

PLOS One